# Dual-Labelled Nanoparticles Inform on the Stability of Fluorescent Labels In Vivo

**DOI:** 10.3390/pharmaceutics15030769

**Published:** 2023-02-25

**Authors:** Sabrina Roussel, Philippe Grenier, Valérie Chénard, Nicolas Bertrand

**Affiliations:** Faculty of Pharmacy, CHU de Quebec Research Center, Université Laval, 2705 Laurier Blvd, Québec, QC G1V 4G2, Canada

**Keywords:** nanomedicines, nanoparticles, fluorophore conjugate, nanoparticle stability, pharmacokinetic, biodistribution

## Abstract

Fluorescent labelling is commonly used to monitor the biodistribution of nanomedicines. However, meaningful interpretation of the results requires that the fluorescent label remains attached to the nanomedicine. In this work, we explore the stability of three fluorophores (BODIPY650, Cyanine 5 and AZ647) attached to polymeric hydrophobic biodegradable anchors. Using dual-labelled poly(ethylene glycol)-*b-*poly(lactic acid) (PEG-PLA) nanoparticles that are both radioactive and fluorescent, we investigated how the properties of the fluorophores impact the stability of the labelling in vitro and in vivo. Results suggest that the more hydrophilic dye (AZ647) is released faster from nanoparticles, and that this instability results in misinterpretation of in vivo data. While hydrophobic dyes are likely more suitable to track nanoparticles in biological environments, quenching of the fluorescence inside the nanoparticles can also introduce artefacts. Altogether, this work raises awareness about the importance of stable labelling methods when investigating the biological fate of nanomedicines.

## 1. Introduction

Nanomedicines can improve the stability and solubility of encapsulated cargos, promote transport across membranes and prolong circulation times to increase safety and efficacy [1]. Nanomedicines prepared with biocompatible polymers are attractive due to their ease of functionalization and the possible customization of their drug release profiles [2,3]. With all new nanomaterial-containing drug products, regulatory agencies require characterizing the intrinsic pharmacological properties of excipients [4]. Examples of important parameters are pharmacokinetics, biodistribution and excretion studies that provide information on potential toxicity considerations upon human use [5]. As such, a key issue for the successful clinical translation of polymeric nanoparticles is a strong understanding of how they interact with biological systems.

Most organic polymers do not have intrinsic spectroscopic properties that allow their detection in biological matrices. Hence, many early-stage pharmacology studies rely on fluorescent labelling to characterize the biological fate of nanomedicines. Fluorescence spectroscopy, with or without microscopy, is a powerful tool to study biological mechanisms, cell internalization and drug release [6,7]. However, a challenge with polymer nanomedicines is the premature release of fluorescent labels upon interaction with biological media [8]. This translates into imprecise characterization of their distribution in vivo. Chen et al. have shown that very hydrophobic fluorescent probes (DiIC_18_ and DiOC_18_) are quickly exchanged from polymer micelles to surrounding proteins and cells, both in vitro [9] and in vivo [10]. Other groups have documented similar exchanges of dyes to biological components from polymeric [11] and non-polymeric nanomedicines [12,13]. This has important implications because it means that, once released, the fluorescent signal is not necessarily representative of the distribution of the nanomedicine. Furthermore, encapsulation of dyes within nanoparticles can lead to fluorescence quenching that can be reverted once the dye is released—hence a loss of linearity in the fluorescence signal [8,14]. In fact, increase in dye fluorescence upon release from nanomedicines is routinely used as a method to monitor the stability of vesicular systems [15]. Specific to poly(ethylene glycol)-*co*-poly(lactic-*co*-glycolic acid) (PEG-PLGA) nanoparticles, Meng et al., showed that dequenching and signal saturation of fluorescence probes can significantly impact the results of nanomedicine biodistribution studies [14]. In an interesting investigation, de Oliveira et al. thoroughly compared the ability of different dyes to remain entrapped within poly(ethylene glycol)-*co*-poly(lactic acid) (PEG-PLA) nanoparticles, after 20 min incubation in plasma [11]. They conclude that hydrophobic molecules (log *p* > 7) and conjugates where the fluorophore is covalently coupled to PLA remain encapsulated in particles more efficiently than other dyes.

To avoid premature release from nanoparticles, the covalent attachment of fluorescent probes to polymers has been used with xanthene core derivatives (e.g., some Alexa Fluors) [16], cyanines [17,18,19], and difluoroboron complexes (e.g., BODIPY) [8,20]. Typically, this labelling is performed by conjugating the terminal end-group of polymers to reactive probes and incorporating these fluorescent conjugates to the nanoparticle preparation. In theory, this approach allows the formulation of labelled nanoparticles (NPs) that are more stable under physiological conditions. However, in the case of PEG-PLGA or PEG-PLA nanoparticles, the hydrophobic polymeric anchors are biodegradable. Here, we wanted to verify whether the properties of the molecules conjugated to biodegradable polymers affected the stability of the labelling process in vivo.

To do so, we used radioactivity as a quantitative reference. The radiolabelling of polymeric NPs can be achieved by chelating radioisotopes, indirect surface labelling with a prosthetic group, radiochemical doping, encapsulation or isotope exchange [21]. This generally allows a quantitative mass balance in preclinical and clinical study with limited effects on the molecular structure or the properties of the polymers [22]. Long-lived β-emitters such has tritium [^3^H] and carbon-14 [^14^C] offer specificity, and long-term stability [23,24]. These isotopes are used as references for mass balance studies in the International Council for Harmonisation of Technical Requirements for Pharmaceuticals for Human Use (ICH) guidelines [25,26]. In our lab, we have developed expertise in incorporating minimum quantities of radioactive molecules to nanomedicine formulations [27,28,29]. To label PEG-PLGA particles, we use a custom [^14^C]-PLGA polymer where the radioactive isotope is randomly distributed throughout the polyester structure, limiting the effects of hydrolytic chain scission on the release of the radioactive tracer [29].

In the present work, we investigate the in vitro and in vivo stability of nanoparticles that are both radioactive and fluorescent and evaluate the impact of labelling instability on their pharmacokinetic and biodistribution. This study proposes that some fluorophores are more limited than others to monitor the biological fate of polymeric nanomedicines. Altogether, this work offers insight on how the labelling strategy can affect the conclusions of pharmacokinetic/biodistribution studies when evaluating the biological fate of nanomedicines in vivo.

## 2. Materials and Methods

### 2.1. Reagents

Poly(ethylene glycol) methyl ether (CAS: 9004-74-4, average Mn 5000 g/mol), 1,5,7-triazabicyclo [4.4.0] dec-5-ene (CAS: 5807-14-7, TBD) and *N*-tBoc-ethanolamine (CAS: 26690-80-2) were purchased from Sigma Aldrich (St. Louis, MO, USA). [^14^C]-poly (lactic-*co*-glycolic acid) copolymer (average Mn 20,000 g/mol, PLGA) was purchased from Moravek Biochemicals (Brea, CA, USA). Cyanine 5 NHS ester (CAS: 1263093-76-0, Cy5) and BODPIPY 650 NHS ester (CAS: 235439-04-0; 1616842-78-4) were purchased from Lumiprobe (MA, USA). AZDye647 NHS ester was purchased from Fluoroprobes (Scottsdale, AZ, USA). All other chemicals and solvents were purchased from Fisher Scientific (Waltham, MA, USA).

### 2.2. Preparation of the PEG-PLA Polymer

The PEG-PLA polymer was synthesized by ring-opening polymerization (ROP) of D,L-lactide at room temperature. The PEG_5k_ methyl ether (PEG_5k_-OH) was used as an initiator, using TBD as an organic catalyst [30]. In a typical reaction, the PEG_5k_-OH (0.11 mmol) and the D,L-lactide (11.45 mmol) are dried for 2 h under vacuum. Then, both reagents are solubilized in 5.7 mL of anhydrous dichloromethane (final concentration of lactide 2M). Initiation of the polymerization occurs when TBD is introduced (0.1 mmol). After 1 h, the polymerization was quenched by adding a small amount of benzoic acid. The polymer solution was concentrated by rotary evaporation and precipitated twice in cold diethyl ether. ^1^H-NMR (CDCl_3_, 400 MHz): δ 1.4–1.7 ppm (457 H, s, O=C-CH-CH-CH_3_-O-H), 3.6–3.8 (454, s, O-CH_2_-CH_2_-O), 5.1–5.3 (151, s, O=C-CH-CH-CH_3_-O-H).

### 2.3. Preparation of Fluorescent Polymers

A protected PLA_10k_ was synthesized following the previous protocol using the *N*-tBoc-ethanolamine as the initiator. To obtain a PLA terminated with an amine function, deprotection was adapted from the Blondelle and Houghten protocol [31]. Briefly, 400 mg of PLA was solubilized in 1 mL of anhydrous dichloromethane. Then 550 µL of trifluoroacetic acid (TFA) was added to the mixture. The reaction was conducted at 0 °C for 12 min. The polymer solution was precipitated twice in cold diethyl ether. The peptidic coupling of the fluorophores was performed between the PLA-NH_2_ and NHS ester dyes. [32]. Briefly, 100 mg of PLA-NH_2_ was dissolved in 400 µL of anhydrous dichloromethane with a small amount of base (diisopropylethylamine, 0.05 mmol). A solution of 2 mg of the NHS-Dye in 200 µL of anhydrous dichloromethane was added to the polymer mixture and reacted, protected from light, at room temperature for 18 h. Purification was performed by precipitation in cold diethyl ether four times to remove the non-reacted dye. The polydispersity and the degree of degradation of the polymers were assessed by conventional size exclusion chromatography (SEC, sometimes known as gel permeation chromatography) in THF (using 2 × 30 cm, Agilent Resipore columns), relative to monodisperse polystyrene standards (InfinityLab EasiCal Polystyrene standards, Agilent, Santa Clara, USA). The chromatogram can be found in Appendix A. ^1^H-NMR (CDCl_3_, 400 MHz): δ 1.4 ppm (9H, s, CH_3_-C-O-C=O), 1.4–1.7 (532H, s, O=C-CH-CH-CH_3_-O-H), 5.0–5.3 (165, s, O=C-CH-CH-CH_3_-O-H).

### 2.4. Dual-Labelled Nanoparticles

The nanoparticles were prepared by nanoprecipitation as described elsewhere, by adding the polymer mixture dropwise to 10 mL of ultrapure water while stirring. [28]. The polymer solution contained 700 µL of the PEG_5k_-PLA_10k_ polymer (10 mg/mL) in acetonitrile, 300 µL of PLA_10k_-Dye (10 mg/mL) in acetonitrile and 30 µL of [^14^C]-PLGA_20k_ copolymer (40 µCi/mL). The resulting nanoparticle suspensions were washed with ultrapure water five times by ultrafiltration (5 min, 3000 g, using Amicon ^®^Ultra-15 100 K cellulose filters). The average diameter (Z-average) and the polydispersity index (PDI) were measured by dynamic light scattering (DLS) at 22 °C with a 173-degree backscatter angle on a Malvern Zetasizer Nano S (Malvern Instruments, Westborough, MA, USA).

### 2.5. Dialysis Experiments

Dialysis was performed as follows: 3 mL of a 1 mg/mL solution of dual-labelled nanoparticles was placed in a 5 cm bag of 100 KDa (Spectrum^TM^ Labs, Biotech, CE, dialysis kit, #08-700-131). The nanoparticles were dialysed against a solution of isotonic phosphate-buffered saline solution pH = 7.2 (potassium phosphate 1.1 mmol/L, NaCl 150 mmol/L, disodium phosphate 3 mmol/L) for seven days. The samples were incubated, under magnetic stirring, protected from light. At specific time points, 100 µL of the retentate (i.e., inside the bag) was collected, and fluorescence and radioactivity were measured. The dialysate was changed at every time point. Experiments were performed in triplicate.

### 2.6. Size Exclusion Chromatography

The size exclusion chromatography was adapted from a method already described elsewhere [33]. Briefly, 200 µL of a 5 mg/mL solution of dual-labelled nanoparticles was loaded on a Sephadex G-75 (1 × 20 cm, DWK Life Sciences Kimble^TM^ FlexColumn, #4204011020, Vineland, NJ, USA) in isotonic phosphate-buffered saline solution pH = 7.2. Fractions of 0.5 mL were collected in microcentrifuge tubes. Aliquots of 100 µL were used to measure the fluorescence with a plate reader (Spectramax i3 molecular device, λ_em/ex_: 645/675 nm) and another 100 µL was used for radioactivity. Size exclusion chromatography process was performed directly after preparation (not shown), and after 1 day of incubation at 37 °C to measure the correlation between fluorescence and radioactivity over a period of 24 h.

### 2.7. Partition Coefficient

Octanol-water partition coefficients were quantified for each free dye using high performance liquid chromatography (HPLC). Each dye was dissolved in methanol (0.8 mg/mL) and a small quantity was dropped into a half/half (*v*/*v*) octanol/water solution. For each phase, 200 µL was taken and lyophilized before being dissolved in 50/50 methanol/water for HPLC analysis. A reverse phase C18 column (Agilent Zorbax Eclipse plus C18, #AG959990-902, USA) in 71/29 methanol/water with 0.15% formic acid was used to measure the area under the curve for each sample. Because PLA is not soluble in octanol, ethyl acetate was used to measure the partition and the degradation product of the dye conjugated to the PLA. Briefly, a solution of 10 mg/mL of each PLA-dye was prepared in acetonitrile. In a conic tube, precisely 1 g of ethyl acetate was weighed and 20 µL of the PLA solution was added. Then, 1 g of water was added to the mix, vortexed and centrifuged for 5 min to allow phase separation. The fluorescence in each phase was measured in triplicate with a plate reader (λ_em/ex_: 645/675 nm). To measure the impact of degradation on the partitioning of fluorescent conjugates, 10 µL of NaoH solution (0.1 to 1 N) was added to 90 µL of the PLA-dye solution in acetonitrile, and stirred for 24 h, protected from the light. Partition in ethyl acetate after degradation was evaluated as described above.

### 2.8. In Vitro Study by Flow Cytometry (Fluorescence Activated Cell Sorting (FACS))

The internalization experiment was conducted in murine macrophage cells (RAW 264.7) using nanoparticles without radioactivity. Twenty-four hours before the experiment, 25,000 cells were plated in 24-well plates in DMEM medium (Wisent Bioproducts, St-Bruno, QC, Canada) containing 10% fetal bovine serum (FBS) and antibiotics. Different quantities of the fluorescent nanoparticles (0–5 µg) were added to the wells and incubated for 2 h. The cell suspension was centrifuged at 300× *g* for 5 min and resuspended in FACS buffer (2% FBS with 1 mM EDTA in PBS) twice before analysis. Flow cytometry experiments were conducted on a BD LSRII (BD Biosciences, Mississauga, ON, Canada) using a 640 nm laser (660/20 filter), and the data was analyzed using FlowJo V10 software (Tree Star Inc., BD, Ashland, OR, USA).

### 2.9. In Vivo Experiment

All animal studies were conducted using approved protocols from Université Laval (Canadian Council on Animal Care Standards and Animal Research: Reporting In Vivo Experiments Guidelines). Pharmacokinetic studies were performed according to a method already published by our laboratory [28]. Healthy animals were housed in a controlled environment (22 °C, 12 h day/night cycle) with ad libitum access to food and water. Six to eight male mice were divided in two groups (radioactivity vs. fluorescence) for each formulation (*n* = 3–4). One hundred uL of dual-labelled formulations (15 mg/mL) was injected intravenously by the subclavian vein or by the intraperitoneal route. Approximately 30–50 µL of blood was collected at various times. Twenty-four hours post-injection, blood was collected by cardiac puncture for a terminal timepoint. Animals were euthanized by cardiac perfusion with 3 mL of isotonic phosphate-buffered saline solution and organs were collected. The blood and organs of three mice were digested at 58 °C (in Solvable^®^, Perkin Elmer, Waltham, MA, USA), bleached with 30% hydrogen peroxide, and assessed by scintillation counting, using Hionic Fluor^®^ (Perkin Elmer, Waltham, MA, USA) as a scintillation cocktail. For mice used for fluorescence analysis, blood samples were centrifuged to isolate the plasma. The plasma was diluted 5-fold and the fluorescence was measured on a 96-well black plate (λ_ex/em_: 645/675 nm). We used a freshly made calibration curve with the nanoparticles in PBS and plasma to assess the percentage of fluorescence at each time point (ranging from 1 to 0.0125 mg/mL of nanoparticles). The organs were analysed by IVIS Lumina II optical system with an excitation wavelength of 640 nm and emission filter for Cy5.5.

### 2.10. Calculation of Pharmacokinetic Parameters

Non-compartmental analysis was used to calculate the pharmacokinetic parameters. The area under the plasma concentration vs. time curve (AUC_0–24h_) was measured by the trapezoidal method from 0 to 24 h. The elimination constant (K_e_) was estimated for IV injection by dividing the clearance (dose/AUC_0-inf_) by the volume of distribution (V_d_= dose/C_0_). The AUC_0-inf_ was extrapolated by dividing the last concentration (C_24h_) by the slope of a semi-logarithmic regression from the last three time points.

### 2.11. Calculation of Discrepancy Indexes (Ic)

The discrepancy index (*Ic*) was adapted from a procedure proposed by Kannan et al. [34] to evaluate physiological pharmacokinetic models that could not use traditional goodness-of-fit statistics. This index uses all experimental points (*N*) and divides the square root of the residual sum of differences between fluorescence (*F*) and a radioactivity (*R*), by the root mean square of radioactivity (i.e., equivalent to the ideal model where fluorescence = radioactivity). The discrepancy index (*Ic*) can be calculated with Equation (1):(1)$$Ic=\frac{\sqrt{\sum_{i=1}^{N}{\left(R_{i}-F_{i}\right)}^{2}/N}}{\sqrt{\sum_{i=1}^{N}{\left(R_{i}\right)}^{2}/N}}$$

This mathematical equation provides a quantitative measure of the degree of discrepancy between experimental values and a model. Low *Ic* values (<0.2) represent good correlation, while increasing values represent a decrease in correlation.

## 3. Results and Discussion

### 3.1. Synthesis and Characterization of the PLA-Dyes

Most dyes used to track nanomaterials in vivo fluoresce in the NIR-I window (600–1000 nm). In this work, we have chosen BODIPY 650 (BD650), Cyanine 5 (Cy5) and AZDye^TM^ 647 (AZ647), three dyes with similar fluorescence spectra (Figure 1A). All fluorophores have high extinction coefficients and quantum yields that make them suitable for detection in biological samples [35]. The first dye is a difluoroboron complex, while the others are carbocyanine derivatives. AZ647, which is structurally similar to Alexa Fluor^®^ 647, has four sulfonate groups that significantly increase its solubility in aqueous environments [20]. The difference in polarity can be appreciated by comparing the partitioning of the dye between octanol and water: the logP values are 2, 1.1 and −1.7 for BD650, Cy5 and AZ647, respectively (Figure 1B).

All fluorophores were purchased with an *N*-hydroxysuccinimide (NHS)-activated carboxylic acid, to allow conjugation to the polymer. To conjugate the dyes, we prepared an amine-terminated poly(lactic acid) (PLA) polymer by ring-opening polymerization at room temperature using TBD as an organic catalyst (Figure 1C) [36]. *N*-tert-butyloxycarbonyl-ethanolamine (*N*-tBoc-ethanolamine) served as an initiator that could be deprotected with minimal hydrolysis of the polyester (Appendix A), to yield the terminal primary amine [37]. In this study, a single polymer with a molecular weight of ~10,000 g·mol^−1^ was used to conjugate all fluorophores. Conjugation of the PLA to the NHS-esters was conducted at room temperature to yield BD650-PLA_10k_, Cy5-PLA_10k_ and AZ647-PLA_10k_ after four rounds of precipitation in cold diethyl ether.

High-molecular-weight PLA is hydrophobic, but slowly degrades via hydrolysis [38]. Pivotal work by Tae Gwan Park in the 1990s [39] has shown that oligomers with molecular weights around 1050 g·mol^−1^ become water soluble and can be released from matrix systems into aqueous environments. Hence, we investigated how hydrolysis affected the hydrophobicity of the fluorescent PLA conjugates. In a model experiment, we accelerated the degradation of polymer conjugates by incubation with increasing concentrations of NaOH in water for 24 h. Figure 1D presents the partitioning of fluorescent molecules between water and ethyl acetate after degradation. Here, ethyl acetate was selected because PLA was insoluble in octanol. These results show that the hydrophobicity of Cy5 and AZ647 conjugates decreases as the polymer degrades, while fluorescent conjugates of BD650 remain highly hydrophobic. Because ester bonds within the polymer are more labile than the terminal amide linkage, we presume fluorescent molecules remain conjugated to oligomers of PLA that are small enough to be water soluble (PLA ≤ 1050 g·mol^−1^) or self-assemble into water-dispersible polymolecular colloids. Of note, while this experiment allows monitoring the partition of fluorescent molecules in the organic and water phase, it was not designed to evaluate if incubation in alkaline solutions had a deleterious effect on the fluorophore structure.

### 3.2. Preparation and Characterization of NPs

Combinatorial synthesis allows the preparation of nanoparticles using mixtures of polymers [29]. In the past, we employed the approach to reproducibly synthetize radioactive [28,29] or fluorescent nanoparticles [27,40]. In the present work, we combined PEG-PLA, fluorescent PLA and [^14^C]-PLGA to prepare nanoparticles that are both fluorescent and radioactive. A 70:30 weight ratio of PEG-PLA to fluorophore-PLA was used for all experiments, with the radiolabel used in trace amounts (i.e., <0.1 wt%). This minute amount of radioactive polymer does not affect the properties of the nanoparticles. Nanoparticles were washed by ultrafiltration five times to remove non-encapsulated labels and solvent [41]. No fluorescence was detected in the filtrate, indicating that the fluorophores were adequately encapsulated in the particles. Due to interference between the fluorophore and the laser from the DLS instrument (λ = 633 nm) [42], samples had to be bleached by sunlight exposure for 4–6 h before measurements were taken. Experiments conducted with non-fluorescent nanoparticles showed that this incubation time did not alter the size of PEG-PLA nanoparticles. The diameter and zeta potential of the nanoparticle was similar for all formulations (diameter ca. 100 nm, PDI ≈ 0.1) (Figure 2). This diameter was also confirmed by nanoparticle tracking analysis using an instrument equipped with a 520 nm laser. Transmission electron microscopy (TEM) of all the fluorescent formulations also showed similar sizes and distributions (Appendix A). Importantly, all formulations showed comparable amounts of radiolabelling (ca. 90,000 dpm per mg of polymer) and were fluorescent (Figure 2B). However, calibration curves obtained by serial dilutions of the nanoparticle suspension highlighted that Cy5 nanoparticles were approximately 3-fold more fluorescent than their AZ647-labelled counterpart, which in turn were 4-fold more fluorescent than BD650 nanoparticles.

### 3.3. In Vitro Release of Labels from the Nanoparticles

Because hydrolysis of fluorescent PLA conjugates yielded more hydrophilic degradation products (as seen in Figure 1D), we investigated the rate at which radioactive and fluorescent labels were released from nanoparticles (Figure 3A). We compared the relative signals of fluorescence and radioactivity remaining in the nanoparticles by sampling, at different intervals, inside a dialysis bag incubated in PBS. In these conditions, formulations containing BD650 and Cy5 released fluorescent and radioactive labels at similar speeds and the ratios of each signal remained constant over a period of 72 h. In contrast, nanoparticles labelled with AZ647 showed a rapid release of their fluorescent signal: the ratio of fluorescence to radioactivity had decreased by 25% within 6 h, and by as much as 75% after 2 days. We have recently developed a method called size exclusion of radioactive polymers (SERP) that allows the detection of radioactive degradation products released from nanoparticles [33]. Simply put, this method uses Sephadex exclusion columns to separate large nanoparticles from smaller hydrolysis products. To better characterize the stability of fluorescent nanoparticles, we used SERP to investigate the correlation between fluorescent and radioactive degradation products. Figure 3B presents the chromatograms obtained with all three dual-labelled formulations after 24 h of incubation in PBS at 37 °C. Although particles remained relatively stable during that period, some hydrolysed species are indeed released from all formulations. Importantly, the correlation between fluorescence and radioactive signals in the degradation products highlights strong differences between the fluorescent conjugates. For BD650 and Cy5, a certain association is observed between both signals. We used a unit-free correlation index (*Ic*) to quantify how well experimental data matches a model where all fractions would be equally fluorescent and radioactive (i.e., dashed lines) [34,43]. With this index, values close to zero (Ic ≤ 0.2) indicate very good fitting, while higher values represent increasing discrepancies between the model and experimental data. For BD650 and Cy5, *Ic* values are below one (i.e., moderately good fitting), and while products that are both fluorescent and radioactive are present, fractions that contain radioactivity but no fluorescence are also detected. These results suggest that, in these in vitro conditions, fluorescence remains associated with the nanoparticles. For AZ647, we appreciated that most fractions are fluorescent but not radioactive. Consequently, *Ic* reaches a value of 33, which is representative of extremely poor correlation between fluorescence and radioactivity.

Altogether, dialysis and size exclusion experiments show that the stability of the nanoparticle fluorescent labelling differs between fluorophores. The more hydrophobic molecules (i.e., BD650 and Cy5) appear to remain inside the nanoparticles more efficiently than hydrophilic molecules (i.e., AZ647). Because all fluorophores are covalently conjugated to the PLA, the release of the dyes is ascribed to the degradation of the hydrophobic polymer anchor.

We also looked at how dye release could impact cellular internalization experiments by flow cytometry using macrophage-like cells (RAW264.7). When 0–5 µg of NPs was incubated for 2 h with RAW264.7 cells, internalization of nanoparticles labelled with Cy5 was much higher than that observed with AZ647 and BDP650 (Appendix A). Because nanoparticles had similar properties (same size, PEG density, zeta potential), we ascribed these differences to either 1) their distinct fluorescence intensity (with Cy5-NP being more fluorescent, fewer internalization events result in positive cells) or 2) the release of the dye in biological media (released dye is not internalized in the same manner as NP). Snipstad et al. [44] observed that premature release of fluorophores and differences in dye physicochemical properties can modify the cellular uptake of NPs.

### 3.4. Pharmacokinetic Profile

In a last set of experiments, we explored whether the differences between fluorescent conjugates observed in vitro had consequences in vivo. Dual-labelled formulations were freshly prepared and intravenously (IV) injected to healthy Balb/c mice. Due to distinct processing of biological samples, the mice receiving each formulation were divided into two groups: one for fluorescence analysis and the other for radioactivity. In the present experiment, radioactivity was measured in full blood and fluorescence in plasma (i.e., without red blood cells). Hence, to allow comparisons we multiplied the respective concentrations measured by the animal’s total blood volume (i.e., 7% of its total weight) and total plasma volume (i.e., 3.5% of its weight, corresponding to an average haematocrit of 50%) [45]. All values in blood/plasma samples were normalized to the injected dose (%ID) using a freshly prepared calibration curve in plasma for fluorescence, and the total number of disintegrations per minute (DPM) injected for radioactivity [16].

The particles had a diameter of 100 ± 10 nm and a PEG density of around 45 PEG chains per 100 nm^2^. These values were chosen because we previously showed that nanoparticles with PEG densities above 20 PEG chains per 100 nm^2^ had comparable pharmacokinetics [29]. Here again, the pharmacokinetics of the particles were similar, and comparable to the circulation profiles obtained previously [40] irrespective of the fluorescent conjugate encapsulated (Appendix A). The blood exposures (as quantified by the area under the plasma vs. time curve (AUC_0–24h_)) for the BD650, Cy5 and AZ647 formulations were 647 ± 65, 631 ± 28 and 593 ± 67 %ID·h per gram of blood, respectively (Appendix A).

Figure 4A represents the distinct circulation profiles obtained between the radioactive and fluorescent labels after intravenous injection of all formulations. For BD650, Cy5 and AZ647, the AUC_0–24h_ obtained with the dyes were approximately 1.4-, 2-, and 5-fold lower respectively than those measured with radioactivity. For BD650 and Cy5, the concentrations measured at the end of the experiment were 2-fold lower with fluorescence compared to those measured with carbon-14. For AZ647, the fluorescence values measured at 24 h were below the quantitation limit of the calibration curve, but the last measurable concentration was 8-fold lower than the one measured with radioactivity at the same time point (i.e., 12 h). This data supports that premature release of the dye from the nanoparticles, again due to polymer degradation, leads into an underestimation of the nanoparticle circulation times. This is particularly true for AZ647, the most hydrophilic dye. The correlation factor *Ic*, used here to compare the circulation profiles obtained with radioactivity and fluorescence, offers a quantitative way of comparing the discrepancies.

Surprisingly, for BD650, the quantities of fluorescent label measured in the first hour are above the total amount injected (i.e., above 100%). These observations remain true even when concentrations are not multiplied by the total plasma volume. We ascribe this increase in fluorescence to the dequenching of the fluorescent molecule as it is released from the nanoparticles upon contact with plasma proteins. Although the hydrophobic BD650 did not result in perceivable dequenching in the studies conducted above in aqueous buffers, a 1.7-fold increase in fluorescence was observed when the nanoparticles were disrupted by solubilisation in acetonitrile (results confirmed with two independent batches of nanoparticles, Appendix A). Comparatively, this increase in fluorescence was not observable with formulations containing other fluorescent conjugates. This supports that the weaker fluorescence of the BD650 formulation (Figure 2) could be attributed to proximity quenching of the fluorescent conjugate within the nanoparticles. Such phenomenon has been described previously with non-conjugated BODIPY derivatives [8].

To further document the instability of the fluorescent labelling in vivo, we investigated the pharmacokinetics of dual-labelled formulations after intraperitoneal (IP) injections. This extravascular administration route requires an absorption phase that increases interactions between nanoparticles and biological components, affording opportunities to understand what happens to degradation products. We have shown previously that nanoparticles injected IP are absorbed into the systemic circulation [40], but that the peritoneum delays the absorption of larger colloids [37].

Figure 4B presents the results obtained after IP dosing of radioactive formulations containing either BD650 or Cy5. With the radioactive signal, the nanoparticles have a bioavailability of ~70% (67 ± 11% for BD650 and 79 ± 10% for Cy5). This is comparable to our previous data obtained with similar nanoparticles [40]. When comparing the AUC obtained with both signals, the radioactive and fluorescence profiles after IP administration are more similar to each other than they were with IV injections. For both conjugates, this translates into a better correlation between the fluorescence and radioactive signals with IP injections compared to IV (Ic values 0.19 vs. 0.4 and 0.25 vs. 0.37 for BD650 and Cy5, respectively). Because the differences were previously attributed to premature release of the fluorophore from nanoparticles in circulation, it seems illogical that a journey in the peritoneum would protect particles against hydrolytic degradation once they reach the bloodstream. However, pharmacokinetic analysis might propose an explanation for these observations. The bioavailability of the fluorescent signal after IP injection (fluorescent-IP) can be calculated with reference to the radioactive or the fluorescent IV signals (radioactive-IV and fluorescent-IV, respectively). For BD650 conjugates, the bioavailability of fluorescent-IP vs. radioactive-IV is 74 ± 9%, while this parameter is 56 ± 7% for the Cy5 conjugates (Figure 4B). This is similar to the values obtained when comparing the radioactive signals by both injection routes. However, when comparing the fluorescent-IP to the fluorescent-IV signal, the bioavailability becomes ≥ 100% (i.e., 101 ± 12% for BD650 and 117 ± 14% for Cy5). These high values are typical of flip-flop pharmacokinetics, where circulation profiles become more dependent on absorption rates than elimination [46]. In other words, this means that the slow absorption of nanoparticles and their degradation in the peritoneal cavity sustains the blood exposure of fluorescent molecules throughout the pharmacokinetic experiment. Flip-flop pharmacokinetics and the resulting increase in blood exposure has been reported after extravascular administration, notably for IP administration of fluorophores [47], oral administration of polymer nanoparticles [48] and sublingual dosing of chitosan nanoparticles [49]. In the present case, although blood exposures to fluorescent and radioactive signals are similar, this does not necessarily mean that both profiles are representative of the biological fate of nanoparticles.

Finally, we looked at the terminal tissue distribution of the nanoparticles, comparing both labelling methods. For radioactivity analyses, tissues were homogenized and assessed by scintillation counting, while fluorescence in whole organs was quantified by IVIS^®^ (using radiant efficiency and the same imaging parameters). Due to these differences in methods, the values obtained with both labels cannot be directly compared. Nevertheless, Figure 5 shows the relative distribution obtained with both methods after IV administration. Without surprise, 24 h after injection, the nanoparticles had mostly distributed to organs of the mononuclear phagocyte system (MPS), the liver and the spleen, but some signal was also detected in the kidneys. The key discrepancy between both labels is the relative signal found in the kidneys. With radioactivity for all three formulations, the %ID found in the kidneys is comparable to the one measured in the spleen, which in turn contains 2- to 3-fold more carbon-14 than the lungs. With fluorescence, the signal in the kidneys is 3-fold higher than the one measured in the spleen, while the signal in the spleen is similar to the one observed in the lungs for BD650 and AZ647, and 2-fold higher with Cy5. Terminal urine samples were also collected, and a 1.4-fold difference in fluorescence/radioactivity ratio was observed between Cy5 and AZ647 (Appendix A). Altogether, these results provide further support that fluorescent degradation products can be released from the nanoparticles and alter the perceived profile of distribution of the nanoparticles.

## 4. Concluding Remarks

Fluorescence and radioactivity are complementary methods that provide information on the fate of nanomedicines in biological systems. While radioactivity is more quantitative, fluorescence is amenable to whole animal imaging, histology and flow cytometry experiments. Understanding how to develop labelling methods that are truly representative of the nanomedicine location and distribution appears of paramount importance to streamline the preclinical development of novel technologies.

In the present work, we highlight how the physicochemical properties of the dyes can impact the stability of the fluorescent labelling once nanoparticles are injected into animals. Despite covalent attachment to polymer chains, hydrophilic dyes (e.g., AZ647) can be more rapidly released from PEG-PLA nanoparticles upon the degradation of the polyester anchor. In parallel, some dyes (e.g., BD650) can undergo fluorescence dequenching upon disruption of the nanoparticle structure, which leads to overestimation of concentrations. We also show that this instability has different implications in vivo based on the injection route of the nanoparticles.

A strength of this work is that it uses the same amine-terminated PLA for the conjugation of all fluorophores, ensuring that differences are truly ascribed to distinctions between fluorescent molecules, and not to variability in polymer chemistry. It is possible that using longer PLA chains (i.e., more hydrophobic) could minimize the release of fluorophores. Studies using dual-labelled systems reassure that discrepancies between signals truly result from the degradation of fluorescent conjugates (i.e., change in fluorescence-to-radioactivity ratio), and not from variation in the physicochemical properties of the nanoparticles.

In parallel, one of the limitations is that the conjugation of fluorophores to the polymer was not optimized. Although thorough purification indicated that limited free dye remained after conjugation, the final stoichiometric ratio between fluorophores and PLA chains might vary (i.e., the quantity of unreacted NH_2_-PLA chains might differ). Because we kept constant the quantity (mass) of fluorescent conjugates used in the formulation of nanoparticles, this can result in differences in encapsulation (i.e., number of fluorophores per nanoparticle). We observed variations in nanoparticle fluorescence, but it is unclear whether this is due to distinct optical properties between dyes or to differences in encapsulation. We did not evaluate whether altering the intrinsic fluorescence of the formulation would change the results of the pharmacokinetic evaluation. For example, the dequenching phenomenon observed with BD650 could presumably be mitigated by diluting the fluorophore. Future work will investigate these effects.

Overall, this work raises more questions than answers. Nevertheless, it increases awareness that not all fluorophores are equivalent, and that the labelling strategy must be considered carefully when designing biodegradable systems. We believe these nuances contribute to the critical examination of pharmacokinetic and biodistribution data, and help the community better discern how their nanomedicines interact with biological systems. Finally, this increase in understanding will streamline the development of technologies that can positively impact human health.

## Figures and Tables

**Figure 1 pharmaceutics-15-00769-f001:**
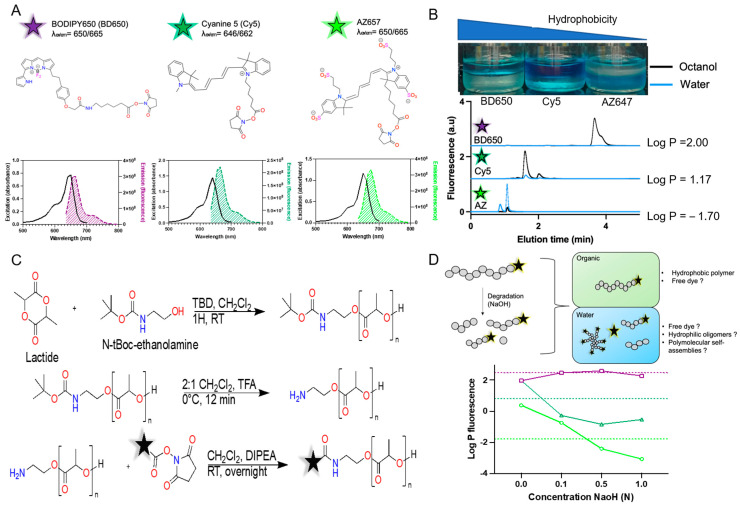
(**A**) Representation of the three dyes and their spectra of absorption/emission. Purple represents BODIPY650 (BD650), dark green Cyanine 5, and light green Alexa Fluor 647 (AZ647). (**B**) The most hydrophilic dye (AZ647) is found mostly in water, compared to Cy5 and BD650. The partition coefficient was established by HPLC by measuring the peak area. Blue line represented the water fraction and black represented the octanol. (**C**) Schematic representation of the polymer synthesis and labelling. The star represented the fluorophore molecules. (**D**) Partitioning of fluorescence between ethyl acetate and water after incubating the polymer in increasing amount of NaOH for 24 h. The dotted line represented the partition coefficient of each dye when not exposed to degradation condition.

**Figure 2 pharmaceutics-15-00769-f002:**
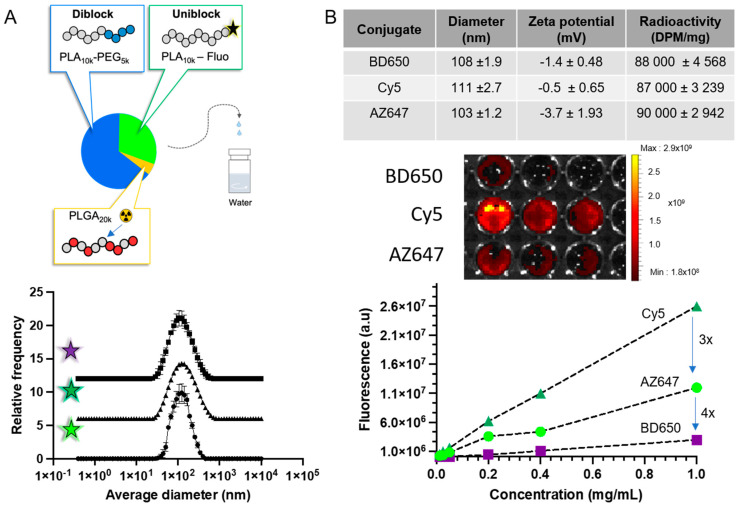
Dual-labelled nanoparticles with similar physicochemical properties were prepared. (**A**) Using the nanoprecipitation method, the polymer self-assembled into core-shell nanoparticles with a size of around 100 nm independently of the fluorophore used. (**B**) Dual-labelled nanoparticles had similar diameters, zeta-potential and radioactivity. The fluorescence was measured by a calibration curve that was used for pharmacokinetic analysis.

**Figure 3 pharmaceutics-15-00769-f003:**
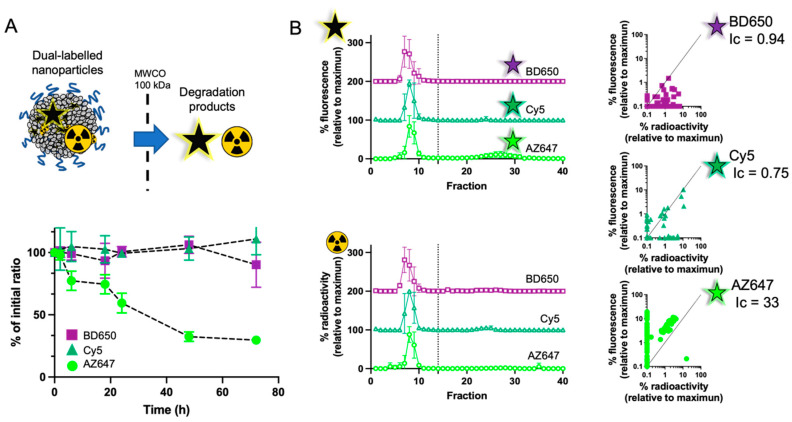
(**A**) Dialysis experiments show the release of fluorescent degradation products from nanoparticles with AZ647. With Cy5 and BD650, the ratio of radioactivity and fluorescence signal remain stable throughout the experiment. (**B**) Using SERP analysis, we show that the degradation products observed with AZ647 are more fluorescent than radioactive. Values represent means ± SD (*n* = 3).

**Figure 4 pharmaceutics-15-00769-f004:**
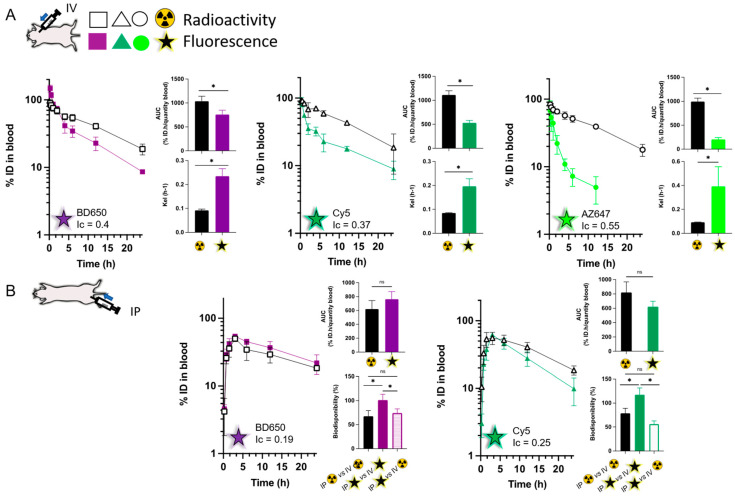
Pharmacokinetic profiles of PEG-PLA nanoparticles after intravenous (**A**) and intraperitoneal (**B**) injections. Open symbol represents radioactivity and full symbol represents fluorescence. Squares represent NPs prepared with BD650, triangles depict NPs with Cy5 and circles represent formulations with AZ647. The data represent the means ± SD (*n* = 3–4). Ns mean the difference is statistically significative (*p* > 0.05), * represented *p* value < 0.05.

**Figure 5 pharmaceutics-15-00769-f005:**
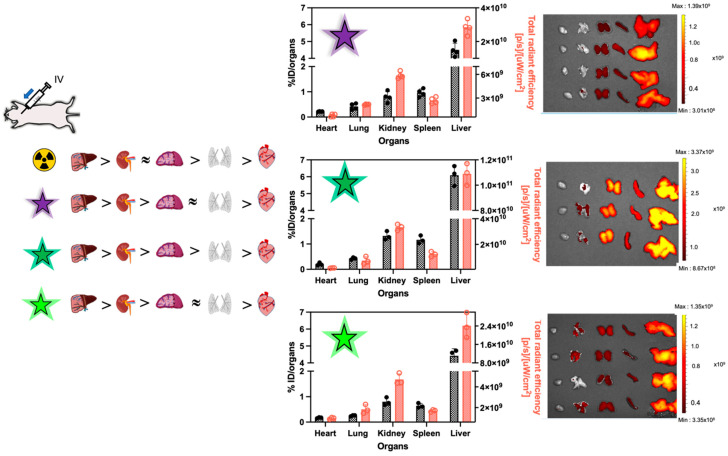
Biodistribution patterns based on radioactivity (black bars, left axis) and fluorescence (red bars, right axis), after intravenous injection. In order, the results are presented as follows: BD650, Cy5 and AZ647 with their IVIS picture related. The organs on the IVIS^®^ picture are listed from left to right as heart, lungs, kidneys, spleen and liver. Data represent means ± SD (*n* = 3–4).

## Data Availability

Data will be made available by authors upon reasonable request.

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
