# Peer review of "Dual-Labelled Nanoparticles Inform on the Stability of Fluorescent Labels In Vivo"

_pharmaceutics, 2023, doi:10.3390/pharmaceutics15030769_

Round 1

Reviewer 1 Report

In this manuscript, Roussel et al, studied the in vitro and in vivo stability of radioactive and fluorescent-labeled PEG-PLA nanoparticles and the impact of labeling instability on nanoparticle pharmacokinetics and biodistribution. This study adds decent knowledge to the existing literature on nanoparticle tracking; however, the authors should address several questions given the limitations of this study.

1) What is the surface charge of these nanoparticles?

2) Fluorescent labeling is frequently used to monitor the cellular-level biodistribution of nanoparticles in vivo. However, the authors did not perform any experiments to show disparities between fluorophores inside the cells.

3) Utilizing only one molecular weight of PLA is a huge limitation in this study. Although the authors do not intend to establish this system as a general phenomenon, it could have a significant impact if it is demonstrated for PEG-PLA nanoparticle types with varying PLA molecular weight.

4) Did the authors perform any urine sample analysis for these studies?

5) Authors should explain why radioactive labeling was performed on PLGA instead of PLA.

6) Measuring the fluorescence of plasma samples is a huge challenge given the interference of biological components. Did the authors inject a high dose of fluorescence material to perform this study?

Author Response

Reviewer #1

  • What is the surface charge of these nanoparticles?

We measured the zeta potential of these nanoparticles (now presented in Figure 2). Our results show neutral zeta potential for all formulations, which is not surprising given the high quantity of PEG on the surface of the nanoparticles.

  • Fluorescent labelling is frequently used to monitor the cellular-level biodistribution of nanoparticles in vivo. However, the authors did not perform any experiments to show disparities between fluorophores inside the cells.

We conducted cellular internalization studies using macrophage-like cells (RAW 264.7) after 2 h incubation. These results are presented in figure Supplementary Figure S6. The manuscript was modified to refer to these results. We observed by flow cytometry that for multiple concentrations, incubation with Cy5-lavelled nanoparticles resulted in larger number of fluorescent-positive cells. Given that the physicochemical properties of the nanoparticles are similar, we ascribed this difference in internalization to either 1) the distinct fluorescence intensity (Cy5-NP being more fluorescent, fewer numbers of internalization events result in positive cells) or 2) the release of the dye in biological media (released dye is not internalized in the same manner as NP).

  • Utilizing only one molecular weight of PLA is a huge limitation in this study. Although the authors do not intend to establish this system as a general phenomenon, it could have a significant impact if it is demonstrated for PEG-PLA nanoparticle types with varying PLA molecular weight.

We agree that using one molecular weight for PLA might be considered a limitation. However in our opinion, it does not change our observations. On the one hand, shorter polymer chains would likely result in similar findings (i.e., release of the conjugated dye after hydrolysis). On the other hand, while it is possible that longer polymer chain would slow down the speed at which the dye is released, it is reasonable to think that it would still happen. Additional work would be necessary to evaluate whether the decelerated release rate would decrease discrepancies between radioactivity and fluorescence in vivo.

  • Did the authors perform any urine sample analysis for these studies?

Urine was collected from the bladder after the mouse sacrifice, and fluorescence and radioactivity were measured. We did appreciate a difference between the fluorescence-to-radioactivity ratio of Cy5 and AD647 nanoparticles. These results are now provided in Supplementary Table S7.  

  • Authors should explain why radioactive labelling was performed on PLGA instead of PLA.

This [14C]-PLGA was custom-synthesized by Moravek Biochemicals. It has been used in our prior work (ref 28 and 29) – this was useful because we know that this polymer offers a good representation of the material’s circulation and biodistribution. The amounts of radiolabeled polymer is very small and does not affect the physicochemical properties of the nanoparticles.  

  • Measuring the fluorescence of plasma samples is a huge challenge given the interference of biological components. Did the authors inject a high dose of fluorescence material to perform this study?

We are aware of the challenges of using fluorescence in vivo – in fact, this paper highlights additional challenges not related to optical properties. Multiple papers highlight that near infrared fluorescence (? = 600-1000 nm) imaging is compatible with in vivo imaging. The dyes used in this paper are also commonly used by other groups, for various applications. All the fluorophores we have used in this study had a similar wavelength (? = 650/665 nm). To remove background noise, the calibration curve was done with plasma. All these factors allowed to minimize the effects of the biological matrix on the signal. The dose injected to the animals is approximately 500 ug of fluorescent polymer.

Reviewer 2 Report

I revised the manuscript Pharmaceutics-2150074 entitled “Dual-labelled nanoparticles for a better understanding of their biodistribution and pharmacokinetic: comparison between fluorescence and radioactivity” by Roussel et al.

In this paper, the authors used dual-labelled poly(ethylene glycol)-b-poly(lactic acid) (PEG-PLA) nanoparticles that are both radioactive and fluorescent and investigated how the properties of the fluorophores impact the stability of the labelling in vitro and in vivo. 

This manuscript presents an important research work and is very well organized, with many assays in vitro and in vivo, and with very high-quality images that prove the quality of this work. 

I leave a few notes (in pdf attachment) that the authors should take into account in order to improve this paper. Also, in figure S2, which not referred to in the manuscript, the authors must remove the remark that is in French. 

Given the above, I am of opinion that this paper should be accepted after the suggested corrections.

Author Response

Reviewer #2

  • I leave a few notes (in pdf attachment) that the authors should take into account in order to improve this paper. Also, in figure S2, which not referred to in the manuscript, the authors must remove the remark that is in French. 

Thank you for your comments. We made the changes suggested to the manuscript.

Reviewer 3 Report

Comments for pharmaceutics-2150074

The paper by Bertrand and coworkers describes “Dual-labelled nanoparticles for a better understanding of their biodistribution and pharmacokinetic: comparison between fluorescence and radioactivity”. The authors presented the details in the manuscript in a good format. Nevertheless, there are still some issues needed to be addressed before publication in this journal.

Comments:

1.      Authors need to provide the GPC chromatograms of polymers.

2.      Authors must be provided with the TEM study of nanoparticles.

3.      Please check for grammar corrections.

Author Response

Reviewer #3

  • Authors need to provide the GPC chromatograms of polymers.

The GPC trace for the polymers are presented in Supplementary Figure S3. We have amended the manuscript to refer to this figure more clearly.

  • Authors must be provided with the TEM study of nanoparticles.

We added TEM analysis for all 3 formulations in Supplementary Figure S4. The formulations used to obtain these pictures were not radioactive for safety reason (i.e., the microscopy facility is not authorized to handle radioactive materials).

  • Please check for grammar corrections.

Grammatical corrections were done.

Round 2

Reviewer 1 Report

The authors adequately responded to the reviewer comments.